# Hepatic inactivation of murine *Surf4* results in marked reduction in plasma cholesterol

Vi T Tang[1,2], Joseph McCormick[2], Bolin Xu[3], Yawei Wang[4], Huan Fang[3], Xiao Wang[3,5], David Siemieniak[2,6], Rami Khoriaty[7,8], Brian T Emmer[7], Xiao-Wei Chen[5], David Ginsburg[2,6,7,9,10]*

[1]Department of Molecular and Integrative Physiology, University of Michigan-Ann Arbor, Ann Arbor, United States; [2]Life Sciences Institute, University of Michigan-Ann Arbor, Ann Arbor, United States; [3]College of Future Technology, Peking University, Beijing, China; [4]Center for Life Sciences, Peking University, Beijing, China; [5]State Key Laboratory of Membrane Biology, Peking University, Beijing, China; [6]Howard Hughes Medical Institute, University of Michigan, Ann Arbor, United States; [7]Department of Internal Medicine, University of Michigan-Ann Arbor, Ann Arbor, United States; [8]Department of Cell and Developmental Biology, University of Michigan, Ann Arbor, United States; [9]Department of Human Genetics, University of Michigan, Ann Arbor, United States; [10]Department of Pediatrics and Communicable Diseases, University of Michigan, Ann Arbor, United States

*For correspondence:
ginsburg@umich.edu

**Competing interest:** The authors declare that no competing interests exist.

**Abstract** PCSK9 negatively regulates low-density lipoprotein receptor (LDLR) abundance on the cell surface, leading to decreased hepatic clearance of LDL particles and increased levels of plasma cholesterol. We previously identified SURF4 as a cargo receptor that facilitates PCSK9 secretion in HEK293T cells (Emmer et al., 2018). Here, we generated hepatic SURF4-deficient mice (*Surf4*[fl/fl] *Alb-Cre*[+]) to investigate the physiologic role of SURF4 in vivo. *Surf4*[fl/fl] *Alb-Cre*[+] mice exhibited normal viability, gross development, and fertility. Plasma PCSK9 levels were reduced by ~60% in *Surf4*[fl/fl] *Alb-Cre*[+] mice, with a corresponding ~50% increase in steady state LDLR protein abundance in the liver, consistent with SURF4 functioning as a cargo receptor for PCSK9. Surprisingly, these mice exhibited a marked reduction in plasma cholesterol and triglyceride levels out of proportion to the partial increase in hepatic LDLR abundance. Detailed characterization of lipoprotein metabolism in these mice instead revealed a severe defect in hepatic lipoprotein secretion, consistent with prior reports of SURF4 also promoting the secretion of apolipoprotein B (APOB). Despite a small increase in liver mass and lipid content, histologic evaluation revealed no evidence of steatohepatitis or fibrosis in *Surf4*[fl/fl] *Alb-Cre*[+] mice. Acute depletion of hepatic SURF4 by CRISPR/Cas9 or liver-targeted siRNA in adult mice confirms these findings. Together, these data support the physiologic significance of SURF4 in the hepatic secretion of PCSK9 and APOB-containing lipoproteins and its potential as a therapeutic target in atherosclerotic cardiovascular diseases.

## Editor's evaluation

Tang et al. demonstrate that the cargo receptor SURF4 is required for the efficient secretion of PCSK9 and apoB from liver. In its absence, blood cholesterol and triglyceride levels are extremely low. These studies carefully and convincingly demonstrate the in vivo function of SURF4 in liver.

## Introduction

An elevated plasma level of low-density lipoprotein (LDL) is a major risk factor for atherosclerotic cardiovascular disease (*Chapman et al., 2011*), which is the leading cause of death worldwide. LDL is derived in the circulation by processing of very-low-density lipoprotein (VLDL) particles, which is synthesized and secreted by the liver. In humans, the major protein component of VLDL is APOB100, which is cotranslationally lipidated in the endoplasmic reticulum (ER) (*Kolovou et al., 2015*). LDL is cleared from circulation by the LDL receptor (LDLR) on cell surfaces. Proprotein convertase subtilisin/ kexin type 9 (PCSK9) is a soluble protein that is secreted by the liver and negatively regulates LDLR abundance by inducing its degradation (*Benjannet et al., 2004*).

Proteins destined for extracellular secretion are transported from the endoplasmic reticulum (ER) to Golgi by COPII coated vesicles/tubules (*Bonifacino and Glick, 2004*; *Palade, 1975*). SEC24 is a key component of the COPII inner coat, which appears to play a primary role in selecting cargo proteins for export from the ER (*Miller et al., 2002*). The mammalian genome encodes 4 paralogs of *Sec24* (*Sec24a-d*) (*Wendeler et al., 2007*). Mice genetically deficient in SEC24A exhibit moderate hypocholesterolemia due to a selective block in PCSK9 secretion from the ER, resulting in an ~50% reduction in plasma PCSK9 levels (*Chen et al., 2013*).

We previously reported a whole genome CRISPR screen in HEK293T cells heterologously expressing PCSK9, identifying Surfeit locus protein 4 (SURF4) as the putative cargo receptor potentially linking PCSK9 within the ER lumen to SEC24A on the cytoplasmic face of the ER membrane (*Emmer et al., 2018*). SURF4 is a 29 kDa protein with multiple transmembrane domains that localizes to the ER and ER-Golgi intermediate compartment (ERGIC) (*Mitrovic et al., 2008*). SURF4 is a homolog of Erv29p, a well-characterized cargo receptor in yeast that mediates the ER to Golgi trafficking of pro-α-mating factor (*Belden and Barlowe, 2001*). The SURF4 ortholog in *C. elegans*, SFT-4, controls the ER export of the yolk protein VIT-2 (*Saegusa et al., 2018*). Recent studies in human cells have also implicated SURF4 in the trafficking of other cargoes, including apolipoprotein B (APOB) (*Emmer et al., 2020*; *Saegusa et al., 2018*; *Wang et al., 2021b*), erythropoietin (EPO) (*Lin et al., 2020*), growth hormone, dentin sialophosphoprotein, and amelogenin (*Yin et al., 2018*). A role for SURF4 in APOB secretion was further supported by a recent study in which acute deletion of hepatic *Surf4* in adult mice caused hypocholesterolemia and a reduction in hepatic lipoprotein secretion (*Wang et al., 2021b*).

To investigate the physiologic significance of SURF4 in the secretion of PCSK9 and other putative cargoes, we previously generated mice with germline deletion of *Surf4*, which resulted in early embryonic lethality (*Emmer et al., 2020*). We now report the generation and characterization of mice with *Surf4* selectively inactivated in the liver by combining a conditional *Surf4* allele (*Surf4^fl^*) with a Cre recombinase expressed under the control of the albumin promoter (*Alb-Cre*). *Surf4^fl/fl^ Alb-Cre^+^* mice exhibit normal development, survival and fertility, with marked plasma hypocholesterolemia associated with a hepatic secretion defect for PCSK9 and APOB-containing lipoproteins without evidence for liver injury. Acute inactivation of hepatic *Surf4* by CRISPR/Cas9 or liver-targeted siRNA in adult mice further confirms these findings and the potential of hepatic SURF4 as a therapeutic target in atherosclerotic cardiovascular disease.

**Table 1.** Genotype distribution of offspring of *Surf4^fl/fl^ Alb-Cre^+^* and *Surf4^fl/+^ Alb-Cre^-^* intercrosses.

| Genotype (Expected) | *Surf4^fl/+^ Alb-Cre^-^* (25%) | *Surf4^fl/+^ Alb-Cre^+^* (25%) | *Surf4^fl/fl^ Alb-Cre^-^* (25%) | *Surf4^fl/fl^ Alb-Cre^+^* (25%) | $p\,(\chi^2)$ |
|---|---|---|---|---|---|
| Mating | ♂ *Surf4^fl/fl^ Alb-Cre^-^* X ♀ *Surf4^fl/+^ Alb-Cre^+^* | | | | |
| | 61 (25.8%) | 71 (30.1%) | 60 (25.4%) | 44 (18.6%) | 0.096 |
| Mating | ♂ *Surf4^fl/+^ Alb Cre^-^* X ♀ *Surf4^fl/fl^ Alb-Cre^+^* | | | | |
| | 8 (18.6%) | 9 (20.9%) | 12 (27.9%) | 14 (32.6%) | 0.549 |

## Results

### Liver-specific deletion of *Surf4* is compatible with normal development and survival in mice

To investigate long term *Surf4* inactivation in hepatocytes in vivo, we generated mice with the *Surf4* gene genetically inactivated specifically in the liver by combining a previously reported conditional *Surf4* allele (in which *Surf4* exon 2 is flanked by loxP sites, denoted *Surf4*[fl]) (*Wang et al., 2021b*) with a *Cre* recombinase transgene under control of the *Albumin* promoter (*Alb-Cre*). *Surf4*[fl/fl] *Alb-Cre*[+] mice

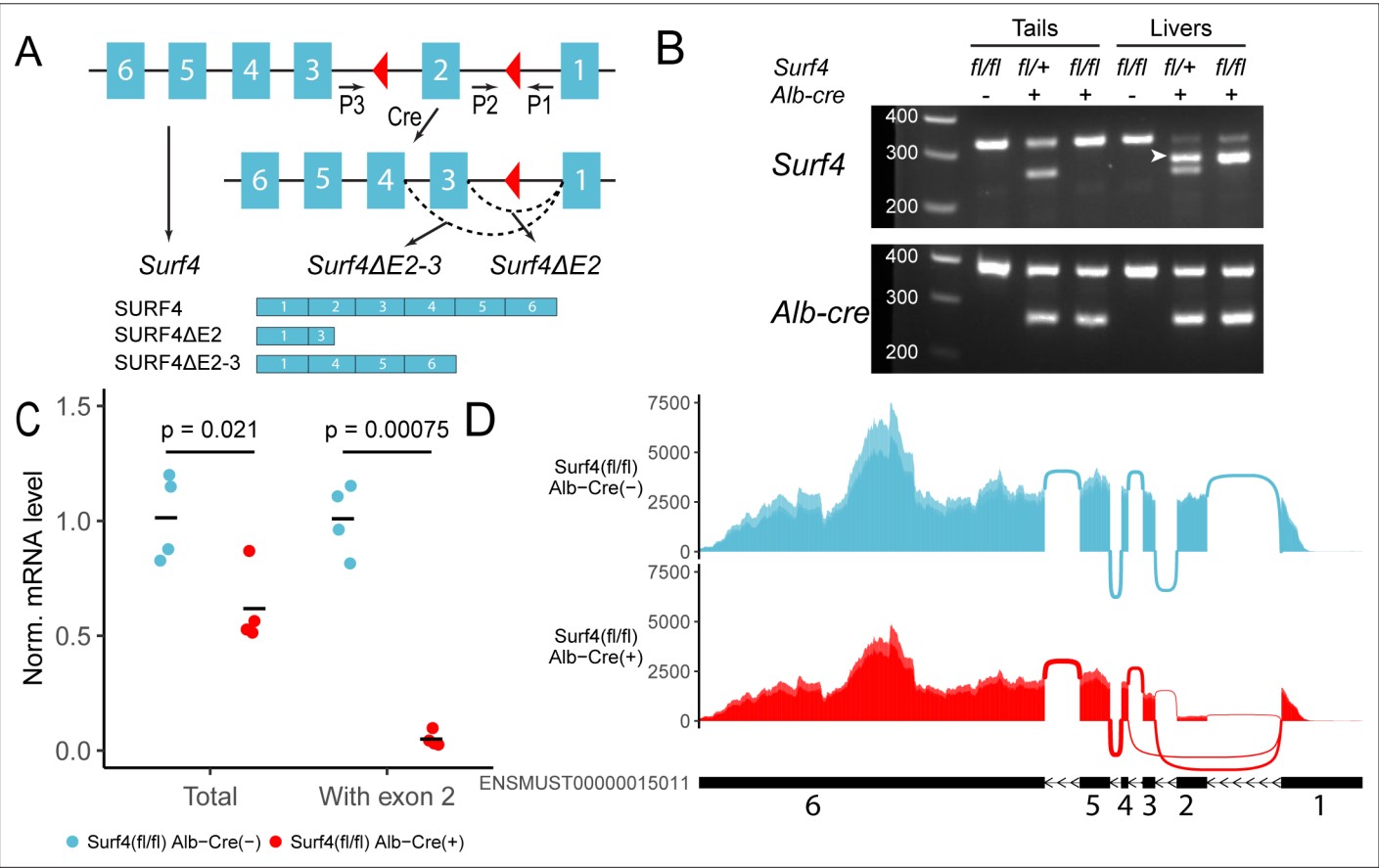

**Figure 1.** Generation of hepatocyte-specific *Surf4* deficient mice. (**A**) Schematic presentation of the *Surf4* conditional allele. Blue rectangles represent exons and black line segments represent introns. Red triangles denote loxP sites. Expression of a *Cre* recombinase leads to excision of exon 2, which results in the generation of a *Surf4* mRNA lacking exon 2 (*Surf4ΔE2*) or both exon 2 and 3 (*Surf4ΔE2-3*). *Surf4ΔE2* mRNA is translated into a truncated SURF4 that is only 22 amino acids in length. *Surf4ΔE2-3* mRNA restores the reading frame, producing an internally truncated protein missing the 88 amino acids encoded by exon 2 and 3. P1, P2, P3 indicate the positions for *Surf4* genotyping primers (*Supplementary file 1*). Dashed arcs represent splicing events. Exons and introns are not drawn to scale. (**B**) Agarose gel electrophoresis of PCR products generated using genomic DNA (gDNA) isolated from mouse tails and livers and primers P1-3 shown in (**A**) (*Figure 1—source data 1*). For *Surf4* genotyping, the wild type allele produces a smaller PCR product whereas the conditional allele produces a larger amplicon. Excision of exon 2 results in the generation of a PCR product of intermediate size (white arrowhead) that is present in gDNA isolated from the livers of *Alb-Cre*[+] mice only. For *Alb-Cre* genotyping, presence of the *Cre* transgene results in a smaller PCR product. The upper band represents the amplification of an internal control (*Supplementary file 1*). (**C**) Quantification of normalized (Norm.) *Surf4* mRNA abundance by quantitative PCR of liver cDNA from control (*Surf4*[fl/fl] *Alb-Cre*[-]) and *Surf4*[fl/fl] *Alb-Cre*[+] mice (n=4 per genotype). Crossbars represent the mean normalized abundance in each group. The denoted p-values were calculated by two-sided Student's t-test. (**D**) Density plots of RNA-seq reads mapping along exons and exon-exon junctions of *Surf4* mRNA. *Surf4*[fl/fl] *Alb-Cre*[+] mice have lower overall read counts due to incomplete nonsense mediated mRNA decay. Arcs between exons represent splicing events and line thickness is proportional to read count. Exact read count for each junction is presented in *Figure 1—figure supplement 1*.

The online version of this article includes the following source data and figure supplement(s) for figure 1:

**Source data 1.** Uncropped and unedited gel shown in *Figure 1B*.

**Figure supplement 1.** Read counts mapping to exon-exon junctions along the *Surf4* transcript based on RNA-sequencing data.

were observed at the expected Mendelian ratio (*Table 1*). Both male and female *Surf4^fl/fl^ Alb-Cre^+^* mice are fertile and produce offspring of the predicted genotypes at expected Mendelian ratios (*Table 1*).

Excision of exon 2 is predicted to result in a frameshift mutation and the generation of a premature termination codon 8 base pairs downstream of the new exon1-3 junction (*Figure 1A*). Analysis of genomic DNA collected from mouse tails and livers of *Surf4^fl/fl^ Alb-Cre^+^* mice demonstrated efficient Cre-mediated excision of *Surf4* exon 2 only in the liver (*Figure 1B*), with the level of exon 2 containing *Surf4* transcripts in *Surf4^fl/fl^ Alb-Cre^+^* mice reduced to ~5% of controls (*Figure 1C*). This residual unexcised *Surf4* mRNA is likely derived from nonhepatocyte cell types in the liver. Quantitative PCR of liver cDNA using primers outside of exon 2 demonstrated a 38% reduction in total *Surf4* mRNA transcript in *Surf4^fl/fl^ Alb-Cre^+^* mice compared to *Surf4^fl/fl^ Alb-Cre^-^* littermates, likely due to nonsense mediated mRNA decay (*Popp and Maquat, 2013*). Analysis of *Surf4* mRNA transcripts by RNA sequencing confirmed the expected reduction of reads spanning the exon 1–2 and exon 2–3 junctions in *Surf4^fl/fl^ Alb-Cre^+^* mice compared to controls (150 and 148 vs 2,623 and 2490 reads, respectively, *Figure 1D* and *Figure 1—figure supplement 1*). Consistent with the qPCR data and incomplete nonsense mediated mRNA decay, we identified 928±51 reads mapping to the exon1-exon3 junction of the *Surf4* mRNA in liver from *Surf4^fl/fl^ Alb-Cre^+^* mice and zero in *Surf4^fl/fl^ Alb-Cre^-^* samples (*Figure 1—figure supplement 1*). This residual exon 2 excised mRNA in *Surf4^fl/fl^ Alb-Cre^+^* liver contains a premature stop codon near the start of the SURF4 coding sequence (codon 23 of 270), which is expected to be translated into a nonfunctional, truncated protein (*Figure 1A*).

We also detected 404±39 reads (none in controls) mapping to the exon1-4 junction of an alternatively spliced *Surf4* mRNA in *Surf4^fl/fl^ Alb-Cre^+^* mice (*Figure 1D*). Exclusion of exon 2 and 3 is predicted to restore the reading frame and result in the production of an internally deleted SURF4, missing ~1/3 of the full length sequence (*Figure 1A*). Though also likely to be nonfunctional, residual activity and/or a dominant-negative effect of this internally deleted SURF4 cannot be excluded.

## Reduced circulating PCSK9 and increased LDLR levels in *Surf4^fl/fl^ Alb-Cre^+^* mice

We previously demonstrated a key role of SURF4 in the efficient trafficking of PCSK9 heterologously expressed in HEK293T cells (*Emmer et al., 2018*). To test the dependence of PCSK9 secretion on SURF4 in vivo, we first examined steady state serum PCSK9 levels in *Surf4^fl/fl^ Alb-Cre^+^* and *Surf4^fl/fl^ Alb-Cre^-^* control mice. Serum PCSK9 levels are reduced by ~60% in *Surf4^fl/fl^ Alb-Cre^+^* mice compared to *Surf4^fl/fl^ Alb-Cre^-^* mice (from 46.0±19.0 ng/mL to 17.8±6.42 ng/mL, *Figure 2A*), although PCSK9 accumulation was not observed in liver lysates (*Figure 2B–C*). Quantitative RT-PCR revealed that *Pcsk9* mRNA levels were also unchanged (*Figure 2—figure supplement 2*), consistent with a defect in PCSK9 protein secretion rather than gene expression as the cause for decreased plasma PCSK9 levels.

In contrast to the above findings, Wang et al reported no change in plasma PCSK9 levels in *Surf4^fl/fl^ Alb-Cre^+^* mice (*Wang et al., 2021a*). To address this issue, and to exclude complex adaptation to hepatic SURF4 deletion induced in utero, we acutely inactivated hepatic *Surf4* in adult mice using a previously reported Cas9 mouse system (*Wang et al., 2021b*). Analyses utilizing three different *Surf4* targeting sgRNAs demonstrated a reproducible ~40% reduction in plasma PCSK9 levels for all mice receiving *Surf4* targeting sgRNA compared to controls (*Figure 2D*), consistent with our findings in *Surf4^fl/fl^ Alb-Cre^+^* mice.

Since PCSK9 is a negative regulator of LDLR, we next quantified LDLR levels in liver lysates collected from control and *Surf4^fl/fl^ Alb-Cre^+^* mice. As shown in *Figure 2E–F*, *Surf4^fl/fl^ Alb-Cre^+^* mice exhibit an ~1.5-fold increase in LDLR abundance in liver lysates compared to controls, consistent with the observed ~60% reduction in circulating PCSK9 level.

## Marked reduction of plasma cholesterol in *Surf4^fl/fl^ Alb-Cre^+^* mice

Humans with heterozygous loss of function mutations in *PCSK9* exhibit an ~28–40% reduction in circulating cholesterol (*Cohen et al., 2005*; *Cohen and Hobbs, 2013*; *Hooper et al., 2007*) with a similar reduction observed in *Pcsk9^+/-^* mice (*Rashid et al., 2005*). Though *Surf4^fl/fl^ Alb-Cre^+^* mice exhibit similar reductions in PCSK9, total serum cholesterol is markedly reduced (from 54.3±15.1 mg/dL in *Surf4^fl/fl^ Alb-Cre^-^* mice to 9.51±2.6 mg/dL in *Surf4^fl/fl^ Alb-Cre^+^* mice) (*Figure 2G*). No change in cholesterol was observed in the limited numbers of mice haploinsufficient for *Surf4* in the liver (*Surf4^fl/+^ Alb-Cre^+^*). Analysis of fractionated pooled sera demonstrated marked reductions in cholesterol and triglyceride

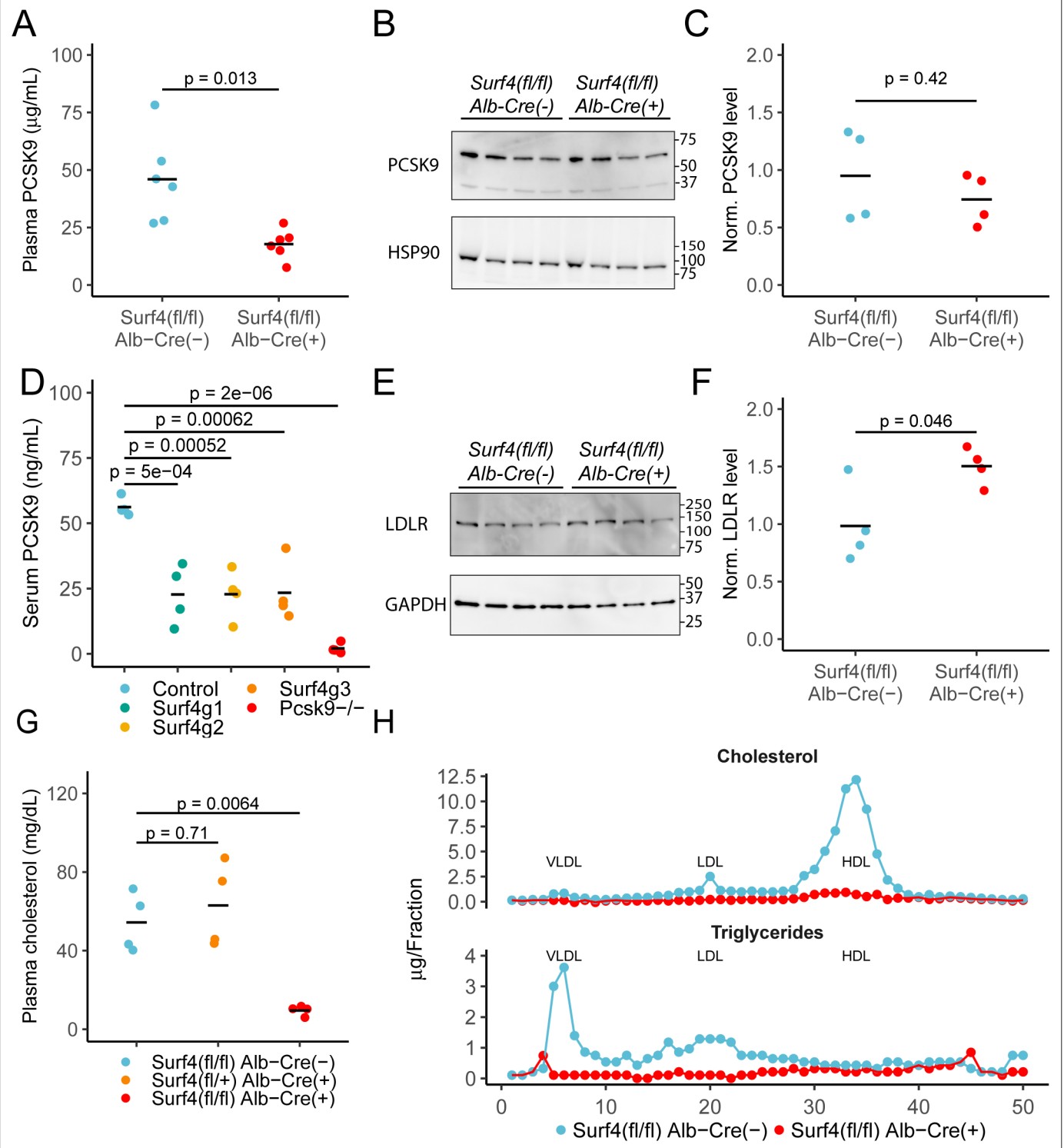

**Figure 2.** Deletion of hepatic *Surf4* results in decreased serum PCSK9 level and profound hypocholesterolemia in mice. (**A**) Serum PCSK9 levels measured by ELISA in *Surf4^fl/fl Alb-Cre^+* mice and *Surf4^fl/fl Alb-Cre^-* littermate controls (n=6 per genotype). (**B**) Immunoblot for PCSK9 and HSP90 (loading control) in liver lysates collected from control and *Surf4^fl/fl Alb-Cre^+* mice (n=4 per genotype) (*Figure 2—source data 1*). (**C**) Quantification of liver PCSK9 levels presented in (**B**) (n=4 per genotype). (**D**) Serum PCSK9 levels in mice in which hepatic *Surf4* was acutely inactivated by CRISPR/Cas9 (n=4 per group). (**E**) Immunoblot of liver lysates collected from control and *Surf4^fl/fl Alb-Cre^+* mice (n=4 per genotype) for LDLR and GAPDH (loading control) (*Figure 2—source data 1*). (**F**) Quantification of liver LDLR levels presented in (**E**). (**G**) Steady-state plasma cholesterol levels in 2 months old control (*Surf4^fl/fl Alb-Cre^-*), heterozygous (*Surf4^fl/+ Alb-Cre^+*), and homozygous (*Surf4^fl/fl Alb-Cre^+*) *Surf4* deleted mice. (**H**) Fractionation of lipoproteins in

*Figure 2 continued on next page*

*Figure 2 continued*

mouse serum by fast protein liquid chromatography (FPLC). Cholesterol and triglyceride levels were measured in each fraction. Each control and *Surf4^fl/fl^ Alb-Cre^+^* sample was pooled from sera of 5 mice. Fractions corresponding to VLDL, LDL, and HDL are annotated. Crossbars represent the mean in all plots. For comparisons between control and *Surf4^fl/fl^ Alb-Cre^+^*, p-values were calculated by two-sided Student's t-test. For comparison between control, heterozygous, and *Surf4^fl/fl^ Alb-Cre^+^ mice*, p-values were obtained by one-way ANOVA test followed by Tukey's post hoc test. Molecular weight markers notated are in kDa.

The online version of this article includes the following source data and figure supplement(s) for figure 2:

**Source data 1.** Uncropped and unedited blots shown in *Figure 2*.

**Figure supplement 1.** Normalized (Norm) *Pcsk9* mRNA levels in *Surf4^fl/fl^ Alb-Cre^-^*.and *Surf4^fl/fl^ Alb-Cre^+^* mice.

**Figure supplement 2.** Hepatic *Surf4* inactiavtion does not affect body mass while hypocholestrolemia is sustained through at least 1 year of age.

content in *Surf4^fl/fl^ Alb-Cre^+^* mice among all three major classes of lipoproteins – very low density lipoprotein (VLDL), low density lipoprotein (LDL), and high density lipoprotein (HDL) (*Figure 2H*). This striking hypocholesterolemia phenotype is sustained through at least 1 year of age (*Figure 2—figure supplement 2*), with no difference in body mass between *Surf4^fl/fl^ Alb-Cre^-^* and *Surf4^fl/fl^ Alb-Cre^+^* mice (*Figure 2—figure supplement 2A* and *Figure 2—figure supplement 2C*).

Consistent with a role of SURF4 in the ER export of APOB-containing lipoproteins (*Wang et al., 2021b*), the mean serum APOB level in fasted *Surf4^fl/fl^ Alb-Cre^+^* mice was 3.88±2.63 mg/mL, a>98% reduction compared to *Surf4^fl/fl^ Alb-Cre^-^* mice (300±157 mg/mL, *Figure 3A*). Western blotting demonstrated a trend towards an accumulation of APOB in the livers of *Surf4^fl/fl^ Alb-Cre^+^* mice compared to littermate controls, predominantly in an endoglycosidase H (endo H) sensitive form (*Figure 3B–C*), indicative of ER retention (*Freeze and Kranz, 2010*).

We next examined hepatic triglyceride secretion in *Surf4^fl/fl^ Alb-Cre^+^* and control mice. For this experiment, mice were fasted to remove intestinal absorption of dietary fat and tissue lipid uptake was blocked by administration of a lipoprotein lipase inhibitor, with liver triglyceride output subsequently monitored by sampling of plasma triglycerides over 24 hours. Following fasting and inhibition of triglyceride hydrolysis, blood glucose levels fell equivalently between *Surf4^fl/fl^ Alb-Cre^+^* and *Surf4^fl/fl^ Alb-Cre^-^* littermates (*Figure 3D*). Although serum cholesterol and triglyceride levels steadily increased in both groups over time, both levels were consistently and significantly lower in *Surf4^fl/fl^ Alb-Cre^+^* mice compared to littermate controls (*Figure 3E–F*). Following an initial decrease in the first hour after lipoprotein lipase inhibition, serum APOB levels steadily rose in control mice (*Figure 3G*). In contrast, serum APOB levels were markedly reduced at baseline in *Surf4^fl/fl^ Alb-Cre^+^* mice, and showed minimal increase after lipoprotein lipase inhibition (*Figure 3G*).

## Intestinal lipid absorption and tissue lipid uptake are unaffected in *Surf4^fl/fl^ Alb-Cre^+^* mice

To assess the potential role of hepatic *Surf4* gene expression on dietary lipid absorption, mice were fed a ^3^H triolein-labelled lipid load following an overnight fast. No significant differences in blood glucose, serum triglycerides, non-esterified fatty acids, or total intestinal uptake of dietary lipids were observed between control and *Surf4^fl/fl^ Alb-Cre^+^* mice (*Figure 4A–D*). Similarly, no significant differences were observed in tissue lipid uptake or fatty acid oxidation between *Surf4^fl/fl^ Alb-Cre^+^* mice and littermate controls (*Figure 4E–F*).

## Loss of liver *Surf4* expression results in mild lipid accumulation but no steatohepatitis or fibrosis

*Surf4^fl/fl^ Alb-Cre^+^* mice exhibited mildly enlarged livers (*Figure 5A*), with a small increase in hepatic fat content and a reduction in lean mass compared to littermate controls (*Figure 5B*). However, no differences were observed in fasting hepatic cholesterol, triglyceride, phospholipid, or nonesterified fatty acid content (*Figure 5C*). Hepatic lipid accumulation can lead to steatohepatitis and liver damage (*Ipsen et al., 2018*). However, at 8–12 week of age, serum albumin, bilirubin, and liver function markers were indistinguishable between *Surf4^fl/fl^ Alb-Cre^+^* and control mice (*Figure 5—figure supplement 1* and *Figure 5D*) and histologic analyses detected no evidence for steatohepatitis or fibrosis (*Figure 5E*). Finally, deep sequencing of liver mRNA identified only limited gene expression changes in response to *Surf4* deletion (*Figure 5F* and *Supplementary file 4*). Significant downregulation was

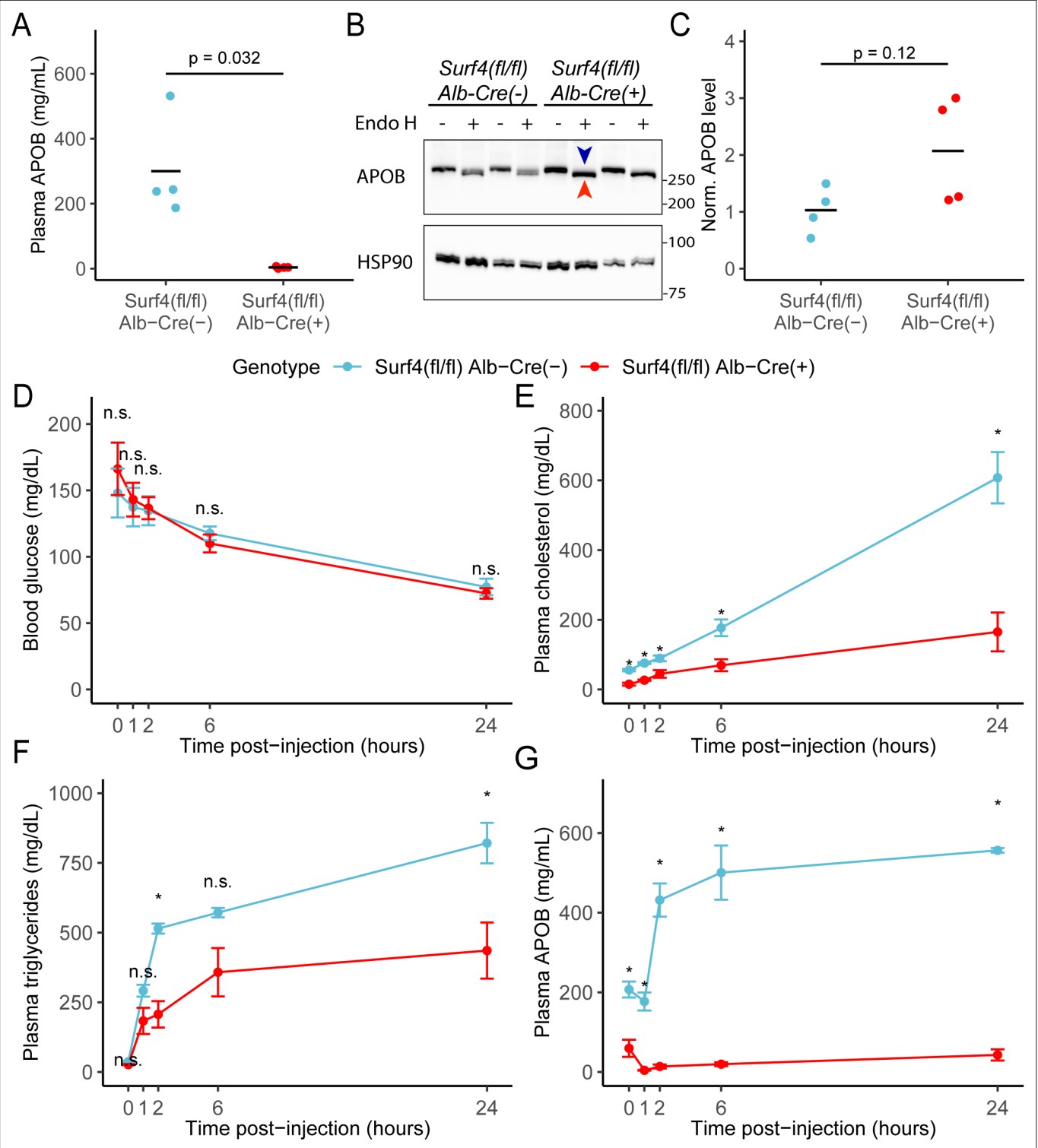

**Figure 3.** Hepatic lipoprotein and APOB secretion defect in *Surf4^{fl/fl} Alb-Cre^+* mice. (**A**) Steady-state serum APOB levels in control and *Surf4^{fl/fl} Alb-Cre^+* mice at 2 months of age (n=4 per genotype). (**B**) Representative immunoblot for APOB and HSP90 in liver lysates with and without endoglycosidase H (endo H) treatment. Proteins in the pre-Golgi compartments are expected to be sensitive to endo H cleavage, resulting in an electrophoretic shift on an immunoblot. Blue arrowhead indicates the endo H resistant band whereas the red arrowhead indicates the endo H sensitive band. Molecular weight markers notated are in kDa. Accumulation of endo H sensitive APOB in the absence of SURF4 suggests accumulation in the ER (*Figure 3—source data 1*). (**C**) Quantification of APOB abundance in control and *Surf4^{fl/fl} Alb-Cre^+* liver lysates, without endo H treatment (n=4 per genotype). For panel A and C, crossbars represent the mean, with statistical significance determined by two-sided Student's t-test. (**D–G**) *Surf4^{fl/fl} Alb-Cre^+* and littermate control

*Figure 3 continued on next page*

*Figure 3 continued*

mice were injected with a lipoprotein lipase inhibitor to block triglyceride hydrolysis. Blood was sampled prior to and following injection over 24 hr and assayed for (**D**) glucose, (**E**) cholesterol, (**F**) triglycerides, and (**G**) APOB levels. Data are presented as mean ± SEM for each time point (n=5 per genotype). Asterisks denote p<0.05 obtained from two-sided Student's t-test with Benjamini-Hochberg adjustment for multiple hypothesis testing, n.s., not significant.

The online version of this article includes the following source data for figure 3:

**Source data 1.** Uncropped and unedited blots shown in *Figure 3*.

observed for several genes involved in fatty acid biosynthesis processes (*Figure 5H*). The most significantly upregulated gene in *Surf4*$^{fl/fl}$ *Alb-Cre*$^+$ mice is *Derl3*, a component of the ER-associated degradation (ERAD) pathway, which could be induced by protein accumulation in the ER of *Surf4*$^{fl/fl}$ *Alb-Cre*$^+$ mice. Genes involved in the unfolded protein response, such as *Ire1*, *Aft6*, and *Perk* (*Hetz et al., 2020*), are not upregulated in *Surf4*$^{fl/fl}$ *Alb-Cre*$^+$ mice (*Figure 5F*).

## Dose-dependent reduction of plasma lipids in response to depletion of SURF4 by siRNA

To confirm the profound hypocholesterolemia with few adverse consequences observed in *Surf4*$^{fl/fl}$ *Alb-Cre*$^+$ mice, and to further explore SURF4 inhibition as a potential therapeutic approach, we next tested depletion of hepatic SURF4 using liver-targeted siRNA in adult mice. Mice were treated with control or *Surf4* targeting siRNA at multiple doses between 0.5 and 4 mg/kg. Mice treated with *Surf4* targeting siRNA demonstrated a dose-dependent reduction of liver *Surf4* mRNA and protein levels (*Figure 6A* and *Figure 6—figure supplement 1*). As expected, plasma PCSK9, cholesterol, triglycerides, APOB, and APOA1 levels were inversely correlated with siRNA dosage, with the highest siRNA dose (4 mg/kg) resulting in cholesterol levels similar to those observed in *Surf4*$^{fl/fl}$ *Alb-Cre*$^+$ mice (*Figure 6B–D* and *Figure 6—figure supplement 1*). Finally, no differences in plasma ALT and AST levels were observed between control and siRNA treated mice, suggesting that siRNA treatment and *Surf4* depletion does not lead to liver injury, even at the highest siRNA dose (*Figure 6E–F*).

## Discussion

We found that embryonic deletion of *Surf4* in hepatocytes results in profound hypocholesterolemia in mice associated with impaired hepatic lipoprotein secretion and normal dietary fat absorption. In addition, we also demonstrated that plasma PCSK9 levels are reduced in *Surf4*$^{fl/fl}$ *Alb-Cre*$^+$ mice. Hepatocyte specific *Surf4* deletion is well tolerated, with only modest increases in hepatic mass and lipid content, and no evidence of hepatic dysfunction or steatohepatitis. Finally, we confirm these findings by siRNA depletion of hepatic SURF4 in adult mice, which leads to reductions of plasma cholesterol, triglycerides, and PCSK9 in a dose dependent manner without apparent deleterious consequences in the liver.

We previously reported that PCSK9 is dependent on SURF4 for efficient secretion in cultured HEK293T cells (*Emmer et al., 2018*). In contrast, Shen et al reported that depletion of SURF4 by siRNA in cultured human hepatocytes leads to increased *Pcsk9* gene expression resulting in increased rather than decreased PCSK9 secretion (*Shen et al., 2020*). The same group also recently reported analysis of *Surf4*$^{fl/fl}$ *Alb-Cre*$^+$ mice, observing no change in plasma PCSK9 levels, in contrast to our findings in a similar genetic model. Our current findings, using three independent mouse models, are consistent with our previous in vitro data and support a physiologic role for SURF4 in facilitating the efficient transport of PCSK9 (as well as APOB) through the secretory pathway. The decrease in plasma PCSK9 in *Surf4*$^{fl/fl}$ *Alb-Cre*$^+$ mice is similar to that observed in SEC24A-deficient mice (*Chen et al., 2013*). Additionally, we also detected a 1.5-fold increase in LDLR level in liver lysates collected from *Surf4*$^{fl/fl}$ *Alb-Cre*$^+$ mice, consistent with the reduction in circulating PCSK9. The increased hepatocyte LDLR levels in *Surf4*$^{fl/fl}$ *Alb-Cre*$^+$ mice are not accompanied by upregulation of *Ldlr* mRNA as measured by RNA-seq analysis, consistent with the increase in LDLR abundance being mediated by PCSK9 activity rather than an increase in gene expression. Furthermore, a recent report by Gomez-Navarro et al independently demonstrated that PCSK9 relies on both SEC24A and SURF4 for secretion and that chemical disruption of SEC24A-SURF4 interaction is sufficient to reduce PCSK9 secretion (*Gomez-Navarro et al., 2022*). Taken together, these data are consistent with the proposed function of SURF4

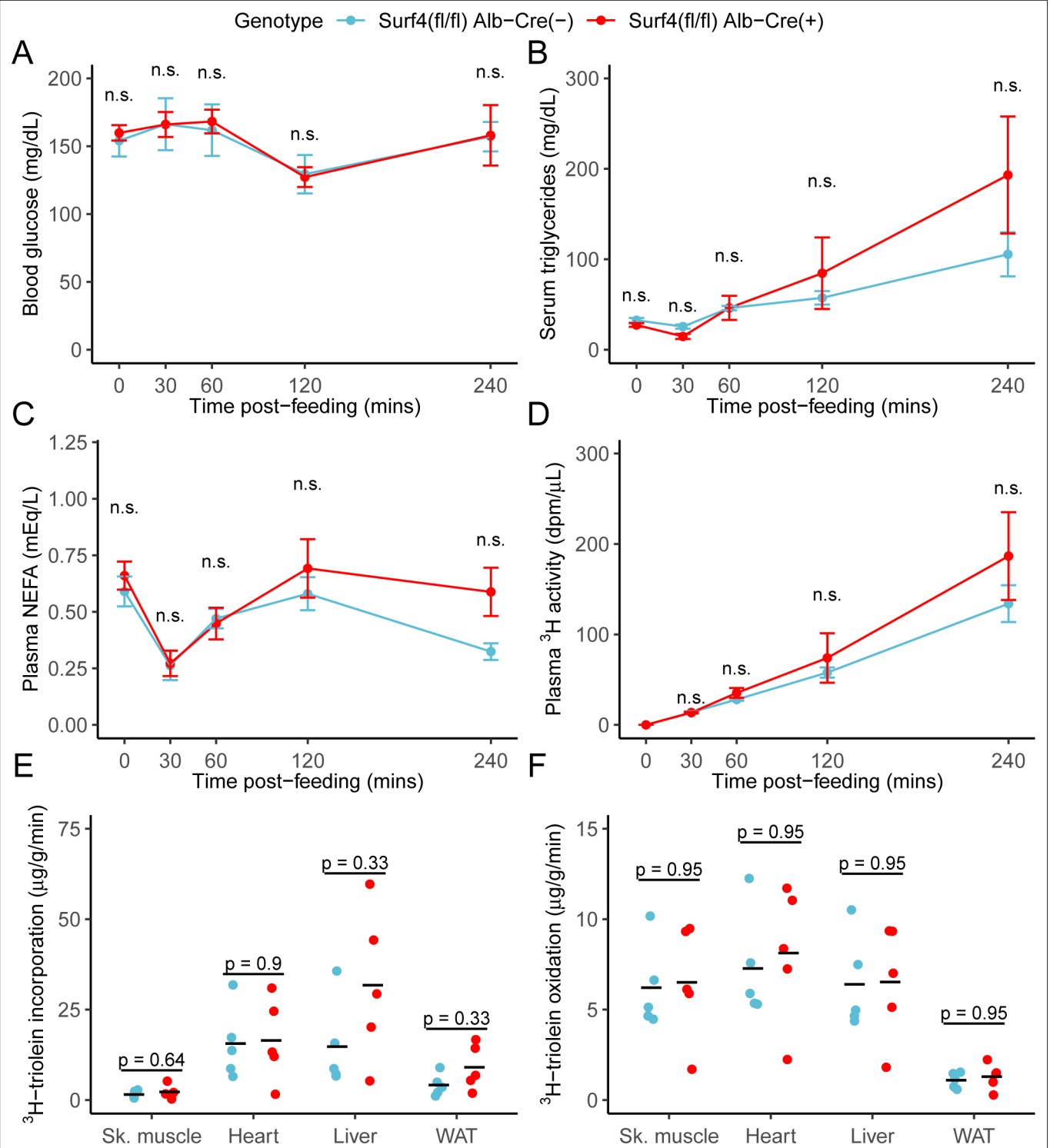

**Figure 4.** Inactivation of hepatic *Surf4* does not impact dietary lipid absorption, incorporation, and oxidation. Mice were administered [3]H-labelled triolein by oral gavage. Blood samples were collected over 4 hr and assayed for (**A**) glucose, (**B**) triglycerides, (**C**) non-esterified fatty acids (NEFA), and (**D**) [3]H radioactivity. Data are presented as mean ± SEM for each time point (n=5 per genotype), n.s., not significant. (**E–F**) Tissues were collected at the 4 hr time point and lipids were extracted by the Folch's method. [3]H radioactivity was measured in the hydrophobic phase, which represents incorporated triolein (**E**) and hydrophilic phase, which represents oxidized triolein (**F**) (n=5 per genotype). All crossbars represent the mean. The denoted p-values were obtained by two-sided Student's t-test with Benjamini-Hochberg adjustment for multiple hypothesis testing.

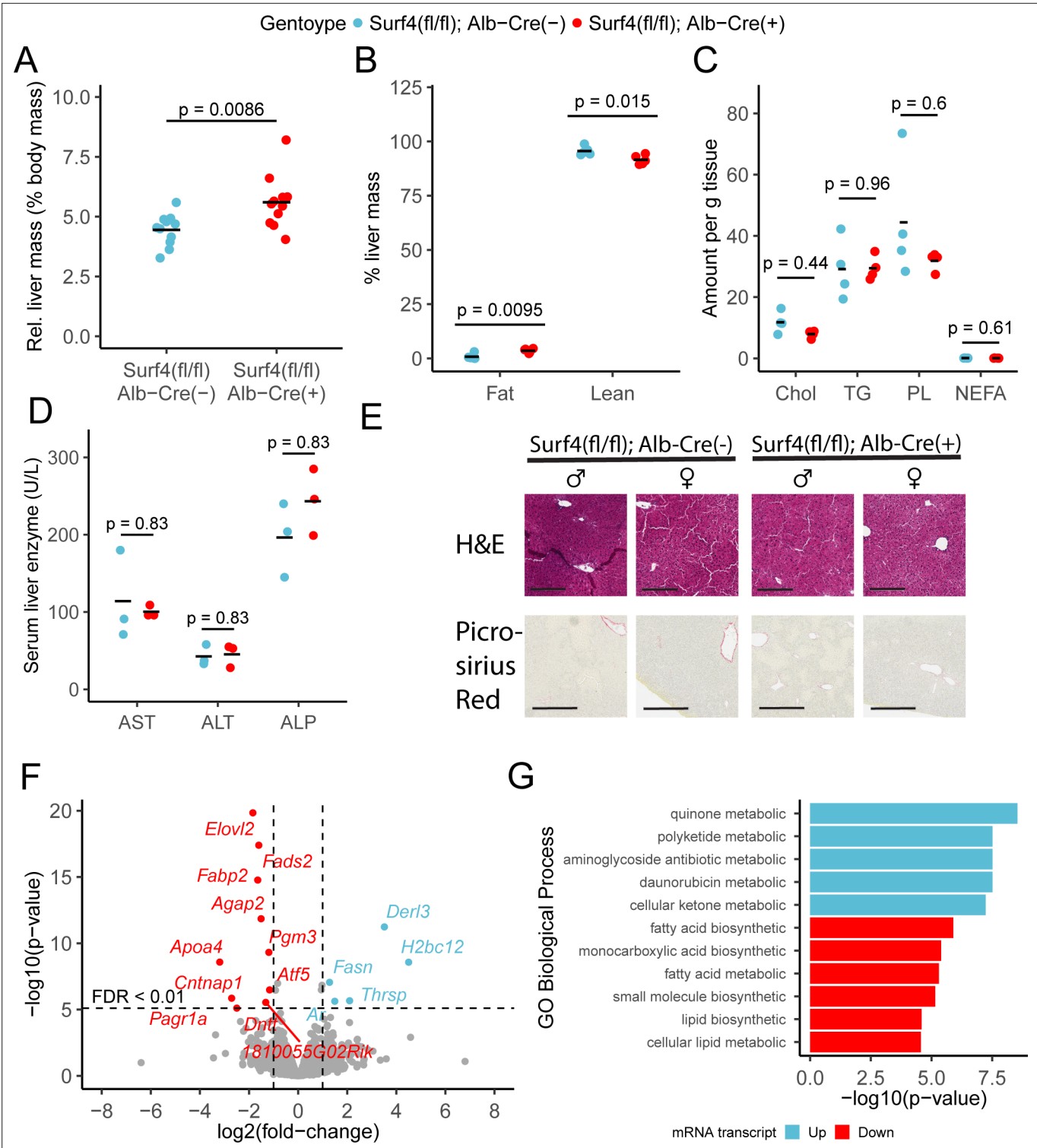

**Figure 5.** Hepatic *Surf4* deletion results in mild hepatomegaly and an increase in liver lipid content, without apparent liver dysfunction or steatohepatitis. (**A**) Relative liver mass in control and *Surf4$^{fl/fl}$ Alb-Cre$^+$* mice presented as percentage of total body mass (n=11 per genotype). (**B**) Relative fat and lean mass in the livers of control and *Surf4$^{fl/fl}$ Alb-Cre$^+$* mice measured by EchoMRI and presented as percentage of liver mass (n=5 per group). (**C**) Levels of cholesterol (Chol, mg/g tissue), triglycerides (TG, mg/g tissue), phospholipids (PL, mg/g tissue), and nonesterified fatty acid (NEFA, mEq/g tissue) in lipids extracted from the livers (n=4 per genotype). (**D**) Serum levels of asparate aminotransferase (AST), alanine transaminase (ALT), and alkaline phosphatase (ALP) (n=4 per genotype). (**E**) Hematoxylin and eosin (H&E) and picrosirius red stained liver sections from control and *Surf4$^{fl/fl}$ Alb-Cre$^+$* mice (n=4 per genotype). Scale bars represent 200 µm in H&E images and 300 µm in picrosirius red images. (**F**) Changes in mRNA

*Figure 5 continued on next page*

*Figure 5 continued*

transcript levels in *Surf4^{fl/fl} Alb-Cre^+* mice compared to littermate controls (n=3 per genotype). Horizontal line represents the p-value above which the false discovery rate (FDR) is less than 0.01. Significantly up (blue) or down (red) regulated transcripts are labelled. (**G**) Significantly overrepresented Gene Ontology (GO) terms for biological processes in up and downregulated gene lists. For panels A-D: All crossbars represent the mean. p-values were obtained by two-sided Student's t-test with Benjamini-Hochberg adjustment for multiple hypothesis testing where appropriate.

The online version of this article includes the following figure supplement(s) for figure 5:

**Figure supplement 1.** Serum (**A**) albumin and (**B**) bilirubin levels in *Surf4^{fl/fl} Alb-Cre^-* and *Surf4^{fl/fl} Alb-Cre^+* mice.

as a cargo receptor linking PCSK9 in the ER lumen to the SEC24A component of the COPII coat on the cytoplasmic face of the ER (*Emmer et al., 2018*). The basis for the discrepancy between our findings and those of B. Wang and colleagues (*Wang et al., 2021a*) is unclear but may be related to differences in mouse genetic or husbandry (*Tang et al., 2017*), or to technical differences in PCSK9 quantification. The profound hypocholesterolemia we observed in *Surf4^{fl/fl} Alb-Cre^+* mice is in agreement with two other studies in which hepatic *Surf4* was acutely inactivated using an AAV/Cas9 mouse system

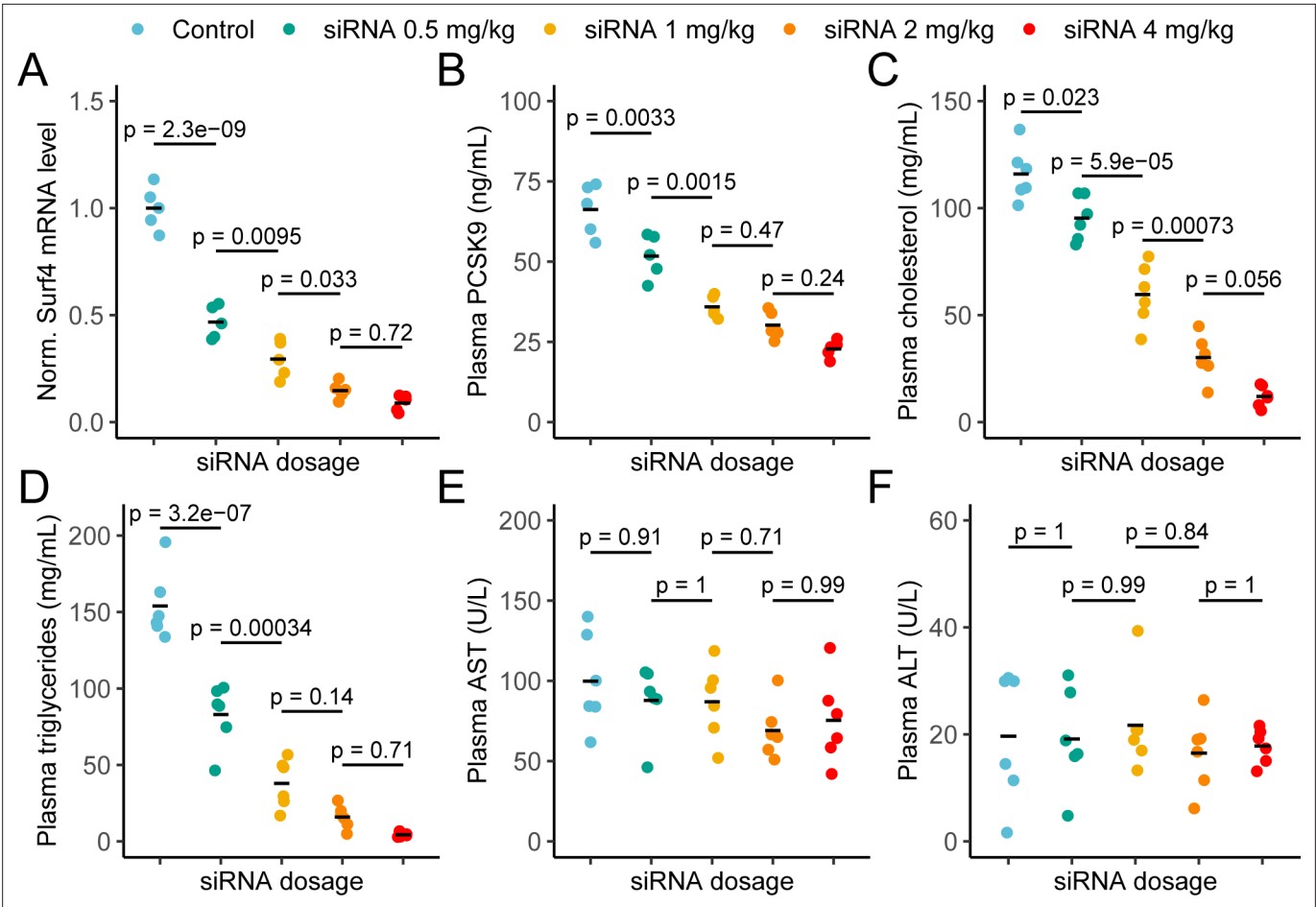

**Figure 6.** Depletion of hepatic *Surf4* by siRNA recapitulates the hypolipidemia seen in *Surf4^{fl/fl} Alb-Cre^+* mice. (**A**) Normalized (Norm.) liver *Surf4* mRNA levels in mice treated with scrambled siRNA (control) or varying concentrations of *Surf4* targeting siRNA (n=5 per group). (**B–D**) Plasma PCSK9, cholesterol, and triglyceride levels in control and siRNA treated mice. (**E–F**) Plasma levels of asparate aminotransferase (AST) and alanine transaminase (ALT) in mice treated with control or increasing doses of *Surf4* targeting siRNA. Statistical significance was computed by one-way ANOVA test followed by Tukey's post hoc test.

The online version of this article includes the following source data and figure supplement(s) for figure 6:

**Figure supplement 1.** Immunoblots of plasma and liver lysates collected from mice treated with control or increasing doses of *Surf4* targeting siRNA (*Figure 6—figure supplement 1—source data 1*).

**Figure supplement 1—source data 1.** Uncropped and unedited blots shown in *Figure 6—figure supplement 1*.

(*Wang et al., 2021b*) or a similar *Surf4*<sup>fl/fl</sup> and *Alb-Cre* model (*Wang et al., 2021a*). We also confirmed this observation in a third model using siRNA-mediated knockdown of *Surf4* transcripts in mouse livers. The decrease in circulating cholesterol is likely due to impaired secretion of APOB containing lipoprotein particles from the liver, and is consistent with multiple previous reports suggesting that APOB is a cargo for SURF4 (*Emmer et al., 2020*; *Saegusa et al., 2018*; *Wang et al., 2021a*; *Wang et al., 2021b*). Despite remarkably low plasma cholesterol levels, *Surf4*<sup>fl/fl</sup> *Alb-Cre*<sup>+</sup> mice exhibit normal growth and fertility compared to littermate controls. Plasma cholesterol is an important precursor for steroid hormone synthesis. However, given that both male and female *Surf4*<sup>fl/fl</sup> *Alb-Cre*<sup>+</sup> mice are fertile, it is unlikely that sex hormone synthesis is significantly perturbed in these mice. Consistent with this conclusion, Chang et al recently reported that even though lipid droplets and cholesterol are depleted in the adrenal glands of *Surf4*<sup>fl/fl</sup> *Alb-Cre*<sup>+</sup> mice, circulating adrenal steroid hormone levels are unchanged under resting and stressed conditions (*Chang et al., 2021*).

Impaired protein secretion could lead to accumulation of proteins in the ER lumen, potentially triggering activation of unfolded protein response pathways and induction of ERAD. Indeed, livers from *Surf4*<sup>fl/fl</sup> *Alb-Cre*<sup>+</sup> mice exhibited upregulation of mRNA for *Derl3*, an ER transmembrane protein that is a functional component of the ERAD complex (*Oda et al., 2006*). While *Derl3* is thought to be a target of the IRE1-XBP1 pathway, we did not detect upregulation of *Ire1* or *Xbp1*, or other ERAD components at the mRNA levels in *Surf4*<sup>fl/fl</sup> *Alb-Cre*<sup>+</sup> mice. Instead upregulation of *Derl3* could be an adaptive response to the protein accumulation in the ER leading to the rapid degradation of these proteins. This can also explain the lack of liver PCSK9 accumulation and a mild increase in liver APOB levels (relative to a significant reduction of plasma levels) in *Surf4*<sup>fl/fl</sup> *Alb-Cre*<sup>+</sup> mice.

Recently, Musunuru et al reported that in vivo CRISPR-mediated base editing of hepatic *PCSK9* leads to an ~60% reduction in plasma cholesterol in cynomolgus monkeys without overt hepato-toxicity (*Musunuru et al., 2021*). Our data suggest that hepatic *Surf4* could be similarly targeted, potentially achieving an even more profound reduction in plasma cholesterol without deleterious consequences. Indeed, it has been shown that inactivation of hepatic *Surf4* is protective against diet-induced atherosclerosis in mice with PCSK9 overexpression (*Wang et al., 2021b*), LDLR deficiency (*Wang et al., 2021a*), and APOE deficiency (*Shen et al., 2022*). Furthermore, polymorphism and mild reduction of *SURF4* expression strongly associate with lower plasma lipid levels and reduced risks of cardiovascular disease in human populations (*Wang et al., 2021b*). Here, we further demonstrate that even more modest reductions in SURF4 induced by siRNA targeting are likely to confer significant lipid-lowering, though such benefits must be weighed against potential toxicity from disrupting the secretion of other SURF4-dependent cargoes.

# Materials and methods

## Key resources table

| Reagent type (species) or resource | Designation | Source or reference | Identifiers | Additional information |
|---|---|---|---|---|
| Strain, strain background (*M. musculus*) | Surf4<sup>fl</sup> mice (C57BL/6 J) | *Wang et al., 2021b* | | |
| Strain, strain background (*M. musculus*) | Albumin-Cre mice (C57BL/6 J) | *Adams et al., 2014* | JAX 003574 | |
| Strain, strain background (*M. musculus*) | Surf4<sup>fl/fl</sup> Alb-Cre<sup>+</sup> mice (C57BL/6 J) | This paper | | |
| Strain, strain background (*M. musculus*) | SpCas9 mice | *Platt et al., 2014* | JAX 026556 | |
| Strain, strain background (*M. musculus*) | C57BL/6 J mice | Jackson lab | JAX 0006640 | |
| Genetic reagent (include species here) | AAV-Cre-sgRNA | This paper | | |

*Continued on next page*

*Continued*

| Reagent type (species) or resource | Designation | Source or reference | Identifiers | Additional information |
|---|---|---|---|---|
| Antibody | Anti-APOB - rabbit polyclonal | Fitzgerald Industries Internationa | 70 R-15771 | WB(1:1000) |
| Antibody | Anti-PCSK9 – rabbit polyclonal | Abcam | ab31762 | WB (1:1000) |
| Antibody | Anti-LDLR – rabbit monoclonal | Abcam | ab52818 | WB (1:1000) |
| Antibody | Anti-HSP90 – rabbit monoclonal | Cell Signaling Technology | 4877 | WB (1:1000) |
| Antibody | Anti-SURF4 – rabbit polyclonal | *Wang et al., 2021b* | | WB (1:1000) |
| Antibody | Anti-APOA1 – rabbit polyclonal | Fitzgerald Industries Internationa | 70 R-15769 | WB (1:1000) |
| Antibody | Anti-Albumin – mouse monoclonal | Proteintech | 66051 | WB (1:1000) |
| Antibody | Anti-Tubulin – rabbit polyclonal | Proteintech | 10094–1-AP | WB (1:1000) |
| Antibody | Anti-GAPDH – rabbit monoclonal | Abcam | ab181602 | WB (1:1000) |
| Sequence-based reagent | Primers | IDT | | Sequences are listed in Supplement File 1 |
| Sequence-based reagent | CRISPR-gRNA | IDT | | Sequences are listed in Supplement File 2 |
| Sequence-based reagent | SURF4-targeting siRNA | This paper | | Sequences are listed in Supplement File 3 |
| Commercial assay or kit | Cholesterol assay kit | SB-1010–225 | Fisher Scientific | |
| Commercial assay or kit | PCSK9 ELSIA | MPC900 | R&D Systems | |
| Commercial assay or kit | APOB ELISA | ab230932 | Abcam | |
| Commercial assay or kit | Triglycerides assay | 10010303 | Cayman Chemical | |
| Commercial assay or kit | RNeasy Plus Mini Kit | 74134 | Qiagen | |
| Commercial assay or kit | Power SYBR Green PCR Master Mix | 4367659 | Invitrogen | |

## Animal care and use

All animal care and use complied with the Principles of Laboratory and Animal Care established by the National Society for Medical Research. Mice were housed in a controlled lighting (12 hr light/dark cycle) and temperature (22 °C) environment and had free access to food (5L0D, LabDiet, St. Louis, MO) and water. All animal protocols in this study have been approved by the Institutional Animal Care and Use Committee (IACUC) of the University of Michigan (protocol number PRO00009304) and the IACUC of Peking University. Both male and female mice were used in this study unless otherwise specified.

## Generation of conditional *Surf4* knockout mice

The generation of mice carrying a conditional *Surf4* allele in which exon 2 of the gene is flanked by 2 loxP sites (*Surf4fl*) has been previously described (*Wang et al., 2021b*). *Surf4fl/+* mice were crossed with mice carrying an *Alb-Cre* transgene (*Adams et al., 2014*) to obtain *Surf4fl/+ Alb-Cre+* mice. These mice were then crossed to *Surf4fl/fl* mice to generate *Surf4fl/fl Alb-Cre+* mice. The *Surf4fl* and *Alb-Cre* alleles were maintained on the C57BL/6 J background by continuous backcrosses to C57BL/6 J mice (0006640, Jackson Laboratory, Bar Harbor ME).

## Genotyping assays

Tail clips were obtained from 2 weeks old mice for genomic DNA isolation and genotyping. PCR was performed using Go-Taq Green Master Mix (Promega, Madison, WI) and resulting products were

resolved by 3% agarose gel electrophoresis. All primers used for genotyping are listed in (*Supplementary file 1*). Those used for genotyping the *Surf4* locus are also depicted in *Figure 1A*. For the *Alb-Cre* transgene, parental mice were genotyped using promoter-specific *Cre* primers and offspring were genotyped with primers that detect the *Cre* transgene (*Supplementary file 1*).

## Blood and tissue collection

Mice were fasted overnight for up to 16 hr prior to sample collection. For non-terminal experiments, blood was collected from the superficial temporal vein using a 4 mm sterile lancet. For terminal experiments, mice were first euthanized by isoflurane inhalation and blood was drawn from the inferior vena cava using a 23 G needle and a 1 ml syringe. Blood was collected into a serum separator tube (365967, BD, Franklin Lakes NJ), allowed to clot at room temperature for at least 10 min, and centrifuged at 15,000 g for 10 min to separate serum. Sera were aliquoted and stored at –80 °C. Liver tissue were collected as previously described (*Emmer et al., 2020*).

## Analysis for sera from *Surf4*^fl/fl^ *Alb-Cre*^+^ mice

Sera were analyzed by a colorimetric assay for total cholesterol (SB-1010–225, Fisher Scientific, Hampton NH) and by ELISAs for PCSK9 (MPC900, R&D Systems, Minneapolis MN) and APOB (ab230932, abcam, Cambridge UK). Serum lipoprotein fractionation assays were performed at the University of Cincinnati Mouse Metabolic Phenotyping Center. Sera were pooled from 5 mice for each genotype and fractionated by fast liquid protein chromatography (FPLC) into 50 fractions. Cholesterol (NC9343696, Fisher, Hampton NH) and triglyceride (TR213, Randox Laboratories, Crumlin UK) content in each fraction were determined using a microliter plate enzyme-based assay. Liver function tests were performed at the University of Michigan In-Vivo Animal Core (IVAC) with sera collected from individual mice using a Liasys analyzer (AMS Alliance).

## Hepatic lipoprotein secretion assay

Hepatic triglyceride secretion assays were performed at the University of Michigan Mouse Metabolic Phenotyping Center as previously described (*Millar et al., 2005*). Blood levels of glucose were measured using a glucometer (Acucheck, Roche, Basel Switzerland) and plasma levels of cholesterol (SB-1010–225, Fisher Scientific, Hampton NH) and triglycerides (10010303, Cayman Chemical, Ann Arbor MI) were determined using colorimetric assay kits. Plasma APOB levels were determined by ELISA (ab230932, abcam, Cambridge UK).

## Oral fat tolerance test and lipid flux assay

Oral fat tolerance test and lipid flux assay were performed at the University of Michigan Mouse Metabolic Phenotyping Center. Mice were fasted overnight and $^3$H triolein-labeled olive oil (0.026 µCi/µl) was given via oral gavage at 5 µl/g of body mass. Blood samples were collected at time 0, 30, 60, 120, and 240 min after the gavage via tail vein bleeding. Plasma levels of triglyceride (10010303, Cayman Chemical, Ann Arbor MI) and non-esterified fatty acid (NEFA-HR (H2), Wako Pure Chemical Industries, Ltd, Richmond VA) were determined using the colorimetric assays. Plasma radioactivity, reported as $^3$H disintegration per minute (dpm), were determined from 2 µl of serum at each time point.

Tissues samples (liver, heart, gastrocnemius muscle, and perigonadal fat) were collected at the 240 min time point, flash frozen in liquid nitrogen and stored at –80 °C. For the liver, tissue composition was measured using an NMR-based analyzer (EchoMRI) immediately upon harvest and prior to freezing. $^3$H-triolein flux was estimated as previously described (*Kusminski et al., 2012*; *Ye et al., 2014*).

## Liver lipid extraction and quantification

Liver lipid extraction and quantification were performed at the University of Cincinnati Mouse Metabolic Phenotyping Center. Lipids were extracted using the Folch's extraction method as previously described (*Folch et al., 1957*). Levels of cholesterol, triglycerides, free fatty acids, and phospholipids in each sample were quantified using specific colorimetric assays.

## Immunoblotting

Lysates were prepared from snap frozen liver tissues and resolved on a 4–20% Tris-glycine gel as previously described (*Emmer et al., 2020*). Immunoblots were probed with antibodies against APOB

(70 R-15771, 1:1000, Fitzgerald Industries International, Acton MA), PCSK9 (ab31762, 1:1000, abcam, Cambridge UK), LDLR (ab52818, 1:1000, abcam, Cambridge UK), HSP90 (4877, 1:1000, Cell Signaling Technology, Danvers MA), SURF4 (*Wang et al., 2021b*), APOA1 (70 R-15769, 1:1000, Fitzgerald Industries International, Acton MA), Albumin (66051, 1:1000, Proteintech, Rosemont, IL), Tubulin (10094–1-AP, 1:1000, Proteintech, Rosemont, IL), and GAPDH (ab181602, 1:5000, abcam, Cambridge UK). For endoglycosidase H assays, 30 µg of lysate was analyzed as previously described using the above antibodies (*Emmer et al., 2018*).

## Histology

Tissue processing, embedding, sectioning, hematoxylin and eosin (H&E), and picrosirius red staining were performed at the University of Michigan In-Vivo Animal Core (IVAC). Slides were reviewed by an investigator blinded to the genotype.

## Analysis of liver mRNA

Liver RNA was isolated from tissue using an RNeasy Plus Mini Kit according to the manufacturer's instructions (74134, Qiagen, Hilden Germany) and reverse transcription was performed using oligo(dT)$_{12\text{-}18}$ primers (18418012, Invitrogen, Waltham MA). Quantitative PCR reactions were performed using Power SYBR Green PCR Master Mix (4367659, Invitrogen, Waltham MA) and primers listed in *Supplementary file 1*. Total *Surf4* mRNA abundance was calculated using data from primers that bind to exon 5 and 6 on the *Surf4* transcript. Abundance of *Surf4* mRNA that contains exon 2 was obtained using primers specific to exon 2 and exon 3 of the transcript. Normalized transcript abundance was calculated by the $2^{-\Delta\Delta Ct}$ method using *Gapdh* and *Rpl37* as housekeeping controls.

Library preparation and next generation sequencing was performed at the University of Michigan Advanced Genomics Core. Demultiplexed fastq files were aligned against the mouse reference genome (GRCm38.92) using STAR (*Dobin et al., 2013*) and quantified with RSEM (*Li and Dewey, 2011*). Differential expression analysis was performed by DESeq2 (*Love et al., 2014*; *Supplementary file 4*). Sequencing coverage for the *Surf4* transcript was analyzed using the ggsashimi package (*Garrido-Martín et al., 2018*). Raw and processed sequencing data have been deposited to GEO (accession number GSE214393).

## Acute inactivation of hepatic *Surf4* in adult mice

Hepatic *Surf4* was selectively inactivated in adult mice by injection of adeno-associated virus (AAV) delivering a hepatocyte-specific *Cre* and a guide RNA targeting *Surf4* or *LacZ* (control) (*Supplementary file 2*) into a Cre-dependent spCas9 knockin mice (*Platt et al., 2014*) as previously described (*Wang et al., 2021b*). Blood samples were collected as previously described (*Wang et al., 2021b*) and plasma PCSK9 concentrations were measured by the commercial kit (CY-8078 of MBL) according to the manufacturer's protocol.

To deplete *Surf4* mRNA, N-acetylgalactosamine (GalNAc) conjugated siRNA oligos are synthesized to ensure liver targeting. SiRNA targeting murine *Surf4* or GalNAc conjugated siRNA with scrambled sequence (siCTL) were injected into 6 weeks old male mice subcutaneously with concentrations indicated in the figures. Seven days after injection, blood samples were collected by tail vein, and then centrifuged at 6000 rpm, 4°C for 10 min to harvest plasma. The plasma PCSK9 concentrations were measured by the commercial kit (CY-8078 of MBL) according to the manufacturer's protocol. SiRNA sequences are listed in *Supplementary file 3*.

## Acknowledgements

This work was supported by NIH grants R35HL135793 (DG), R01HL148333 (RK), R01HL157062 (RK), and K08HL148552 (BTE); the National Key R&D Program grant 2018YFA0506900 (XWC); the National Science Foundation of China grants 91957119, 91954001, 31571213, and 31521062 (XWC); the American Heart Association Predoctoral Fellowship 20PRE35210706 (VTT); the University of Michigan Rackham Predoctoral Fellowship (VTT). DG is a Howard Hughes Medical Institute Investigator. Funding sources were not involved in study design, data collection and interpretation, or the decision to submit the work for publication

# Additional information

## Funding

| Funder | Grant reference number | Author |
|---|---|---|
| American Heart Association | 20PRE35210706 | Vi T Tang |
| National Institutes of Health | R35HL135793 | David Ginsburg |
| National Institutes of Health | R01HL148333 | Rami Khoriaty |
| National Institutes of Health | R01HL157062 | Rami Khoriaty |
| National Institutes of Health | K08HL148552 | Brian T Emmer |
| National Key Research and Development Program of China | 2018YFA0506900 | Xiao-Wei Chen |
| National Science Foundation of China | 91954001 | Xiao-Wei Chen |
| National Science Foundation of China | 31571213 | Xiao-Wei Chen |
| National Science Foundation of China | 31521062 | Xiao-Wei Chen |
| University of Michigan | Rackham Predoctoral Fellowship | Vi T Tang |
| Howard Hughes Medical Institute | | David Ginsburg |

The funders had no role in study design, data collection and interpretation, or the decision to submit the work for publication.

## Author contributions

Vi T Tang, Conceptualization, Formal analysis, Investigation, Visualization, Methodology, Writing – original draft, Writing – review and editing; Joseph McCormick, Investigation; Bolin Xu, Yawei Wang, Huan Fang, Xiao Wang, Investigation, Methodology; David Siemieniak, Software, Formal analysis; Rami Khoriaty, Brian T Emmer, Conceptualization, Supervision, Investigation, Writing – review and editing; Xiao-Wei Chen, Conceptualization, Formal analysis, Supervision, Investigation, Methodology, Writing – review and editing; David Ginsburg, Conceptualization, Formal analysis, Supervision, Funding acquisition, Writing – review and editing

## Author ORCIDs

Vi T Tang ⓘ http://orcid.org/0000-0001-6079-9756
Brian T Emmer ⓘ http://orcid.org/0000-0001-7365-1021
Xiao-Wei Chen ⓘ http://orcid.org/0000-0003-4564-5120
David Ginsburg ⓘ http://orcid.org/0000-0002-6436-8942

## Ethics

All animal care and use complied with the Principles of Laboratory and Animal Care established by the National Society for Medical Research. All animal protocols in this study have been approved by the Institutional Animal Care and Use Committee (IACUC) of the University of Michigan (protocol number PRO00009304) and the IACUC of Peking University.

## Decision letter and Author response

Decision letter https://doi.org/10.7554/eLife.82269.sa1
Author response https://doi.org/10.7554/eLife.82269.sa2

# Additional files

## Supplementary files
- MDAR checklist
- Supplementary file 1. Primer sequences used in this study.
- Supplementary file 2. Guide RNA (gRNA) sequence used for AAV-CRISPR mediated hepatic *Surf4* inactivation.
- Supplementary file 3. siRNA sequences used to target hepatic *Surf4* in this study.
- Supplementary file 4. DESeq2 output for differential gene expression analysis of RNA-seq data.

## Data availability
Sequencing data have been deposited in GEO (accession number GSE214393). All data generated or analyzed during this study are included in the manuscript and supporting files. Source data files have been provided for figure 1, figure 2, figure 3, and figure 6-figure supplement 1.

The following dataset was generated:

| Author(s) | Year | Dataset title | Dataset URL | Database and Identifier |
|---|---|---|---|---|
| Tang VT, McCormick J, Xu B, Wang Y, Fang H, Wang X, Siemieniak D, Khoriaty R, Emmer BT, Chen XW, Ginsburg D | 2022 | Hepatic inactivation of murine Surf4 results in marked reduction in plasma cholesterol | https://www.ncbi.nlm.nih.gov/geo/query/acc.cgi?acc=GSE214393 | NCBI Gene Expression Omnibus, GSE214393 |

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
