## [Editor Report]

Tang et al. demonstrate that the cargo receptor SURF4 is required for the efficient secretion of PCSK9 and apoB from liver. In its absence, blood cholesterol and triglyceride levels are extremely low. These studies carefully and convincingly demonstrate the in vivo function of SURF4 in liver.

---

## [Decision Letter]

**Decision letter after peer review:**

Thank you for submitting your article "Hepatic inactivation of murine *Surf4* results in marked reduction in plasma cholesterol" for consideration by *eLife*. Your article has been reviewed by 3 peer reviewers, including Randy Schekman as Reviewing Editor and Reviewer #3, and the evaluation has been overseen by Anna Akhmanova as the Senior Editor. The following individual involved in review of your submission has agreed to reveal their identity: Charles Barlowe (Reviewer #2);

The reviewers have discussed their reviews with one another, and the Reviewing Editor has drafted this to help you prepare a revised submission. The reviewers had a general concern about the overlap of your findings and those reported by your co-author in a publication last year by X. Wang et al. Nonetheless, you have appropriately cited this work and the new results on PCSK9 secretion and the hepatic accumulation of LDLR in the Surf4 fl/fl Alb-Cre+ strain is considered a valuable addition meriting publication of this report as an Advance on your previous *eLife* publications on this topic.

Essential revisions:

Reviewers #1 and 2 have several recommendations for small changes that you should be able to accommodate with no additional experimental effort.

*Reviewer #1 (Recommendations for the authors):*

Tang et al., in this report investigate the effects of deleting Surf4 in mouse liver. Previously this group has shown that Surf4 functions as a cargo receptor that facilitates the secretion of PCSK9 in cultured cells. Here they have deleted the gene in hepatocytes and find that there is a significant reduction in plasma PCSK9 levels with a resulting increase in LDLR protein and lowering of plasma cholesterol levels. Surf deletion in hepatocytes using albumin-Cre had no deleterious effects in liver. What was found was a 60% reduction in plasma PCSK9 with no change in PCSK9 mRNA levels. These results were confirmed using Cas9 mice in which Surf4 was acutely deleted. Consistent with the known function of PCSK9, the reduction in plasma PCSK9 was associated with a significant increased in liver LDLR protein levels. In addition to dramatically lower plasma cholesterol levels in all lipoprotein fractions, they also find reduced plasma TG levels they show was due to a marked reduction in apoB and TG secretion. Interestingly, there was no defect in intestinal lipid absorption.

1. Page 7. Humans heterozygous for loss-of-function PCSK9 mutations have ~28% reduction in plasma LDL cholesterol.

2. Surf4 deletion actually did not result in lipid accumulation in liver as TG levels were not different that controls. This is interesting given the marked reduction in TG secretion and the fact that apoB or MTTP mutations result in significant hepatic steatosis. Is this due to reduced fatty acid and triglyceride synthesis as suggested possibly by RNA seq? If so-what led to this reduced gene expression? To this reviewer's knowledge, this reduction is not observed in apoB or MTTP knockout mice.

*Reviewer #2 (Recommendations for the authors):*

1. Suggest adding information in the methods section that fully describes housing conditions and diet for mice in this study including light/dark cycle if used and composition of chow diet.

2. Recommend including the full data set of mRNA sequencing data from Figure 5F. Ideally this could be an excel file in the supplement. There may be some changes in mRNA levels that are borderline and would be useful to other investigators. Moreover, in the results and Discussion sections it is mentioned that mRNA levels for genes involved in the UPR and for LDLR expression were not upregulated in Surf4 deficient mice. Presumably this data is contained in the RNA-seq results.

3. The Figure 3 legend on page 28 is confusing. It is stated that "Accumulation of endo H sensitive SURF4 suggests accumulation in the ER." Recommend changing this to "Accumulation of endo H sensitive APOB in in the absence of SURF4 suggests accumulation in the ER."

4. Molecular weight markers should be added to immunoblots shown in Figures 2, 3, and S5. Presumably the blot in Figure 3B is detecting ApoB-100.

5. Figure S5 shows an immunoblot using anti-Surf4 antibodies although the source of these antibodies was not described and suggest including in the methods section.

*Reviewer #3 (Recommendations for the authors):*

Tang et al., have created a Surf4 fl/fl Alb-Cre+ strain of mice that has allowed the evaluation of liver specific, developmental effects of the deletion of the SURF4 gene on the secretion of PCSK9 and of lipoproteins, cholesterol and triglycerides. This study continues a line of investigation initiated in the Ginsburg lab that led to the discovery that SURF4 is the ER sorting receptor for PCSK9, the plasma protein involved in regulating the itinerary of the LDL receptor (Chen, X et al., 2013 and Emmer et al., 2018). This manuscript was submitted as an Advance on the most recent paper in this series where the team has now shown that SURF4 is required for optimal secretion of PCSK9 in vivo. This work challenges a recent publication by B. Wang et al., (2021) where the claim is made that liver specific deletion of SURF has no effect on the secretion of PCSK9. The differences between the strain developed for this work and that created by for the work of B. Wang et al., is not obvious and the explanation offered by Tang et al., for the difference would be hard to check. Nonetheless, Tang et al., compared their results with another approach that was described last year by X. Wang et al., and the outcome was also a reduction in the level of plasma PCSK9. Thus by two different approaches, the authors find a consistent role for SURF4 in the secretion of PCSK9 in vivo.

In another important result, Tang et al., show an increase in the level of the liver LDLR in the SURF4 liver specific knockout. This result conforms to the published observations that PCSK9 enhances the down regulation and turnover of the LDLR in cultured cells.

In other significant results, Tang et al., report a dramatic reduction in plasma cholesterol and triglycerides and an equally impressive decline in the secretion of liver lipoproteins in the null strain.

The results and conclusions are sound and certainly add confidence to the conclusion that SURF4 is important in the secretion of PCSK9 in vivo. On the down side, this work duplicates much of the important work and results of several of the co-authors who published many of these findings last year (X. Wang et al.) Using an acute deletion of SURF4 in the adult liver, X. Wang et al., already reported a dramatic reduction in plasma cholesterol and triglycerides and a substantial block in the secretion of LDL, HDL, VLDL, APOB and APOA1.

---

## [Author Response]

Reviewer #1 (Recommendations for the authors):Tang et al., in this report investigate the effects of deleting Surf4 in mouse liver. Previously this group has shown that Surf4 functions as a cargo receptor that facilitates the secretion of PCSK9 in cultured cells. Here they have deleted the gene in hepatocytes and find that there is a significant reduction in plasma PCSK9 levels with a resulting increase in LDLR protein and lowering of plasma cholesterol levels. Surf deletion in hepatocytes using albumin-Cre had no deleterious effects in liver. What was found was a 60% reduction in plasma PCSK9 with no change in PCSK9 mRNA levels. These results were confirmed using Cas9 mice in which Surf4 was acutely deleted. Consistent with the known function of PCSK9, the reduction in plasma PCSK9 was associated with a significant increased in liver LDLR protein levels. In addition to dramatically lower plasma cholesterol levels in all lipoprotein fractions, they also find reduced plasma TG levels they show was due to a marked reduction in apoB and TG secretion. Interestingly, there was no defect in intestinal lipid absorption.1. Page 7. Humans heterozygous for loss-of-function PCSK9 mutations have ~28% reduction in plasma LDL cholesterol.

We thank the reviewer for raising this point. There are multiple figures reported in the literature and we have edited the text to 28-40% reduction in plasma LDL cholesterol and cited a recent review paper by experts in the field (Cohen and Hobbs, 2013).

2. Surf4 deletion actually did not result in lipid accumulation in liver as TG levels were not different that controls. This is interesting given the marked reduction in TG secretion and the fact that apoB or MTTP mutations result in significant hepatic steatosis. Is this due to reduced fatty acid and triglyceride synthesis as suggested possibly by RNA seq? If so-what led to this reduced gene expression? To this reviewer's knowledge, this reduction is not observed in apoB or MTTP knockout mice.

While levels of triglycerides, along with cholesterol, phospholipids, and free fatty acid were not depleted in Surf4(fl/fl) Alb-Cre(+) mice (figure 5C), we did notice an slight increase in liver lipid content in these mice as demonstrated by two independent methods (EchoMRI and Folch’s lipid extraction from the tissue). At the moment, we do not know the identities of these lipid species. A comprehensive lipidomic study would be needed to answer this question, which we hope the reviewer and editors would agree is beyond the scope of the current manuscript. Additionally, we now note in the text that the mice used in this study are relatively young (3-4 months old) and it is possible that hepatic steatosis might develop in older mice.

Reviewer #2 (Recommendations for the authors):1. Suggest adding information in the methods section that fully describes housing conditions and diet for mice in this study including light/dark cycle if used and composition of chow diet.

We thank the reviewer for raising this point. We have added housing conditions and diet information to the text (page 12).

2. Recommend including the full data set of mRNA sequencing data from Figure 5F. Ideally this could be an excel file in the supplement. There may be some changes in mRNA levels that are borderline and would be useful to other investigators. Moreover, in the results and Discussion sections it is mentioned that mRNA levels for genes involved in the UPR and for LDLR expression were not upregulated in Surf4 deficient mice. Presumably this data is contained in the RNA-seq results.

We have included an excel file with the full results for the RNA-seq data. Additionally, we have now deposited the raw sequencing files to the GEO.

3. The Figure 3 legend on page 28 is confusing. It is stated that "Accumulation of endo H sensitive SURF4 suggests accumulation in the ER." Recommend changing this to "Accumulation of endo H sensitive APOB in in the absence of SURF4 suggests accumulation in the ER."

We thank the reviewer for catching this error. We have edited the figure legend text as suggested (page 29).

4. Molecular weight markers should be added to immunoblots shown in Figures 2, 3, and S5. Presumably the blot in Figure 3B is detecting ApoB-100.

We have added molecular weight markers to the blots in Figure 2 and 3 as suggested. The antibodies used in Figure S5 have previously been validated so MW markers were not included when the blots were run.

5. Figure S5 shows an immunoblot using anti-Surf4 antibodies although the source of these antibodies was not described and suggest including in the methods section.

The antibody against SURF4 was previously generated as described in X. Wang et al., 2021. We have included this information and the sources for other antibodies used in the method section (page 15 and 16).

Reviewer #3 (Recommendations for the authors):Tang et al., have created a Surf4 fl/fl Alb-Cre+ strain of mice that has allowed the evaluation of liver specific, developmental effects of the deletion of the SURF4 gene on the secretion of PCSK9 and of lipoproteins, cholesterol and triglycerides. This study continues a line of investigation initiated in the Ginsburg lab that led to the discovery that SURF4 is the ER sorting receptor for PCSK9, the plasma protein involved in regulating the itinerary of the LDL receptor (Chen, X et al., 2013 and Emmer et al., 2018). This manuscript was submitted as an Advance on the most recent paper in this series where the team has now shown that SURF4 is required for optimal secretion of PCSK9 in vivo. This work challenges a recent publication by B. Wang et al., (2021) where the claim is made that liver specific deletion of SURF has no effect on the secretion of PCSK9. The differences between the strain developed for this work and that created by for the work of B. Wang et al., is not obvious and the explanation offered by Tang et al., for the difference would be hard to check. Nonetheless, Tang et al., compared their results with another approach that was described last year by X. Wang et al., and the outcome was also a reduction in the level of plasma PCSK9. Thus by two different approaches, the authors find a consistent role for SURF4 in the secretion of PCSK9 in vivo.In another important result, Tang et al., show an increase in the level of the liver LDLR in the SURF4 liver specific knockout. This result conforms to the published observations that PCSK9 enhances the down regulation and turnover of the LDLR in cultured cells.In other significant results, Tang et al., report a dramatic reduction in plasma cholesterol and triglycerides and an equally impressive decline in the secretion of liver lipoproteins in the null strain.The results and conclusions are sound and certainly add confidence to the conclusion that SURF4 is important in the secretion of PCSK9 in vivo. On the down side, this work duplicates much of the important work and results of several of the co-authors who published many of these findings last year (X. Wang et al.) Using an acute deletion of SURF4 in the adult liver, X. Wang et al., already reported a dramatic reduction in plasma cholesterol and triglycerides and a substantial block in the secretion of LDL, HDL, VLDL, APOB and APOA1.

We thank the reviewer for the critiques. We acknowledge that the hypocholesterolemia phenotype in hepatic SURF4 deficient mice and the roles of SURF4 in the secretions of several lipoproteins have been reported by some of the co-authors (X. Wang et al). Our work add confidence to these findings using a different genetic model (embryonic liver-specific deletion). Further, while the findings in siRNA mediated hepatic *Surf4* knockdown mice were similar to those seen in mice with acute liver SURF4 deletion; they were different approaches. Together, we demonstrated the roles of SURF4 in the liver using three separate mouse models, which we hope provides important validation for the previous studies, as well as valuable new data.